# Correlation between Ground $^{222}$Rn and $^{226}$Ra and Long-Term Risk Assessment at the at the Bauxite Bearing Area of Fongo-Tongo, Western Cameroon

Léonard Boris Djeufack [1,2], Guillaume Samuel Bineng [1], Oumar Bobbo Modibo [2], Joseph Emmanuel Ndjana Nkoulou II [2] and Saïdou [1,2,*]



1 Nuclear Physics Laboratory, Faculty of Science, University of Yaoundé I, Yaoundé P.O. Box 812, Cameroon
2 Research Centre for Nuclear Science and Technology, Institute of Geological and Mining Research (IRGM), Yaoundé P.O. Box 4110, Cameroon
* Correspondence: saidous2002@yahoo.fr; Tel.: +237-674-174-473

**Simple Summary:** This paper presents a study of radioactivity in soil that included an assessment of radiological risk parameters and long-term health risks from exposure to naturally occurring radionuclides in soil at the bauxite-bearing area of Fongo-Tongo in Western Cameroon. The radionuclides measured in the soil had concentration values above the recommended limits. However, the total long-term excess risk at the site decreased progressively over the years, and the maximum value of $8.58 \times 10^{-3}$ was obtained at T = 0 years. In addition, the external pathway is the largest contributor to the total excess risk assessed inside the building. The maximum risk value for this pathway, which is $2.33 \times 10^{-2}$, was obtained at T = 30 years before decreasing sharply thereafter.

**Abstract:** The aim of the current work was to study natural radioactivity in soil and the correlation between $^{222}$Rn and $^{226}$Ra in the ground and to assess the onsite and indoor long-term excess cancer risk at the bauxite bearing area of Fongo-Tongo in Western Cameroon. $^{222}$Rn was measured in the ground at a depth of one meter, using Markus 10 detector. $^{226}$Ra, $^{232}$Th, and $^{40}$K activity concentrations were measured in soil by two techniques, in situ and laboratory gamma spectrometry. The mean values of $^{222}$Rn concentrations in the ground were $69 \pm 18$ kBqm$^{-3}$ for Fongo-Tongo and $82 \pm 34$ kBq m$^{-3}$ for the locality of Dschang, respectively. The mean values of $^{226}$Ra, $^{232}$Th, and $^{40}$K activity concentrations obtained with in situ gamma spectrometry were $129 \pm 22$, $205 \pm 61$, and $224 \pm 39$ Bq kg$^{-1}$ for $^{226}$Ra, $^{232}$Th, and $^{40}$K, respectively, and those obtained by laboratory gamma spectrometry were $129 \pm 23$, $184 \pm 54$, and $237 \pm 44$ Bq kg$^{-1}$, respectively. A strong correlation between $^{222}$Rn and $^{226}$Ra activity concentrations determined by in situ and laboratory measurements ($R^2 = 0.86$ and $0.88$, respectively) was found. In addition, it is shown that the total excess cancer risk has a maximum value of $8.6 \times 10^{-3}$ at T = 0 year and decreases progressively in the long term. It is also shown that $^{226}$Ra makes a major contribution, i.e., above 70%, to the total excess cancer risk.

**Keywords:** radium-226; radon-222; life excess cancer risk

## 1. Introduction

Areas with high mining potential generally represent a very interesting field for environmental monitoring before, during, and after mining. In the case of the sites hosting not-yet-exploited ore deposits, activities related to exploration led to the transfer of soil from underground to the ground surface. This action could lead to the environmental pollution by natural radioactive materials, increasing the exposure level of inhabitants to natural radiation. Moreover, human exposure to natural radiation sources is ubiquitous and inescapable. Radionuclides in the earth's crust vary from one environment to another, depending on the soil and geological profile [1]. The content and type of radioelement depend, therefore, on the bedrock [2–4]. A long exposure to the natural radionuclides

($^{226}$Ra, $^{232}$Th, and $^{40}$K) is mainly responsible for some cancers and sometimes for the effects of genetic mutations. They constitute a real threat to human health [5–7].

Many investigations on natural radioactivity have been made in the world [8–10]. The investigation conducted in Cameroon revealed the occurrence of high levels of radioactivity in some specific areas of the country [11–13]. These high radioactivity levels are more localized in areas with uranium, thorium, and bauxite mining potential. It is the case of the natural radioactivity measurements made in Poli and Lolodorf, Douala, Fongo-Tongo, Dschang, and Ngaoundal [13–15]. They revealed high $^{238}$U, $^{232}$Th, and $^{40}$K activity concentrations in soil compared to their corresponding world levels, as well as $^{222}$Rn and $^{220}$Rn concentrations in dwellings above the WHO reference level [12,16–18]. These mentioned studies showed that the $^{222}$Rn level in homes depends considerably on the type of architecture, geological structure, and mineralogical composition of soil of the area [16,19]. A good correlation between $^{238}$U and $^{232}$Th activity concentrations in soil with $^{222}$Rn and $^{220}$Rn in dwellings was found in the areas, respectively [20,21].

However, these studies have not specifically examined the correlation that may exist between $^{222}$Rn and $^{226}$Ra activity concentrations in soil. $^{222}$Rn is a direct progeny of $^{226}$Ra [1]. Therefore, its concentration in soil should be proportional to that of the direct parent, $^{226}$Ra. $^{222}$Rn measurement was performed by using a MARKUS 10 detector to a depth of 1 m in the ground. The determination of the activity concentrations of $^{226}$Ra, $^{232}$Th, and $^{40}$K in soil was performed with a NaI (Tl) gamma spectrometer. In addition, the $^{222}$Rn and $^{226}$Ra activity concentrations determined by in situ and laboratory measurements are strongly correlated, and these correlation coefficients were determined. Radiological parameters (AEED, Ra$_{eq}$, H$_{in}$, H$_{ex}$, ELCR, I$_\gamma$, and I$_\alpha$) were determined to assess the level of public exposure to natural radioactivity in the area, and a map of the distribution of $^{226}$Ra and $^{222}$Rn concentrations in soil was established.

## 2. Materials and Methods

### 2.1. Study Areas

The area is located on the mountainous chain region of the Western Cameroon, specifically at the southwestern flanks of the Bamboutos Mountains [22]. The climate of the area is sub-equatorial, Cameroonian type, cold and humid, characterized by a long rainy season (March–November) and a short dry season (December–February). The average temperature and rainfall in the area are 22.5 °C and 1364.4 mm over the year, respectively [23,24]. The soils are Andic type, ferrallitic, trachytic, granitic, and basaltic [22,24,25].

This area is underlain by an extensive and thick loose mantle developed on trachyte and generally forms a differentiated geological profile, including the presence of deposits formed by new bauxite minerals; it was discovered in 1957 by BUMIFOM prospectors [26,27]. This locality is one of the main bauxite deposit sites in the western region of Cameroon. Its potential is estimated at 45 million tons and is a part of the major geological reserves of Cameroon [26,28]. Figure 1 shows a geological map of the study area.

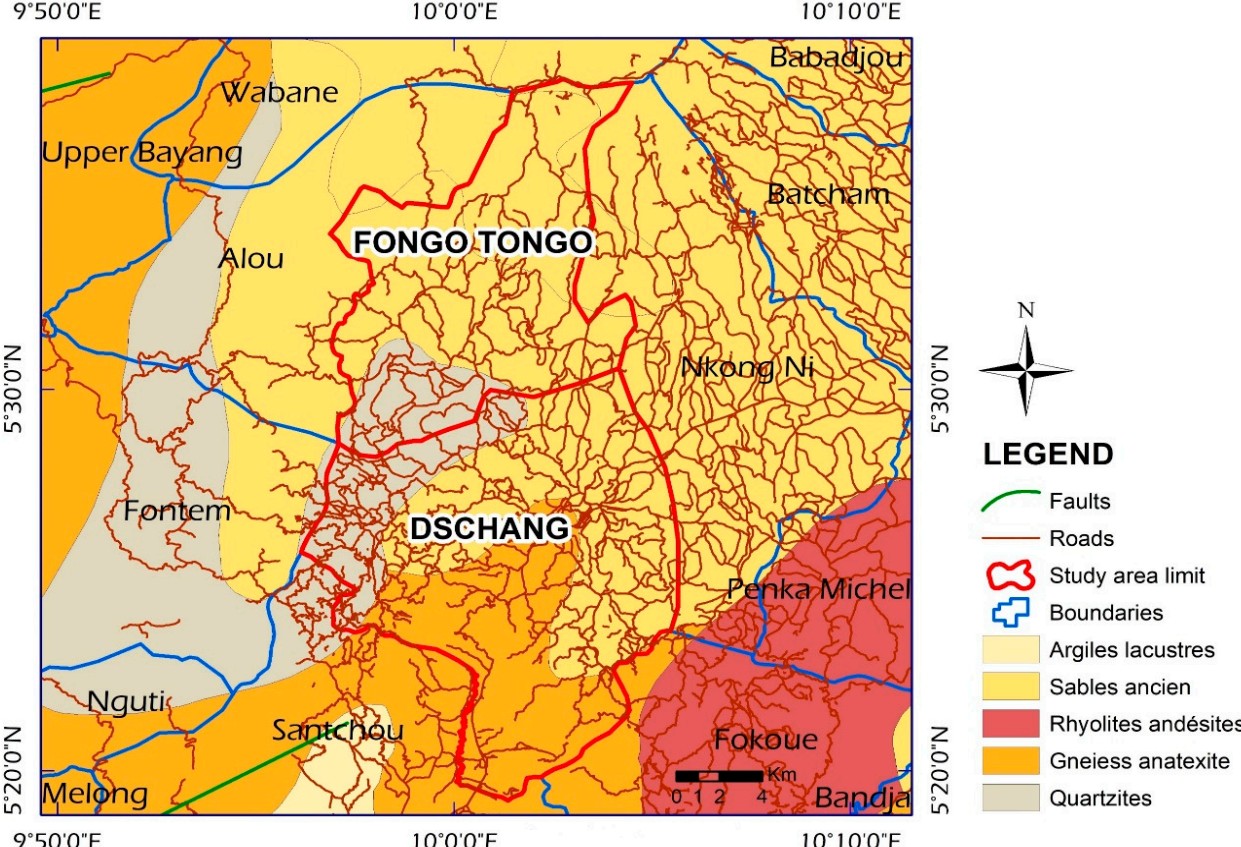

**Figure 1.** Soil characteristics map of the area.

*2.2. Natural Radioactivity Measurements*

2.2.1. Radioactivity Measurements in Laboratory

A total of twenty-seven soil samples (fifteen in Dschang and towel in Fongo-Tongo) were randomly collected for a depth between 0 and 5 cm. The samples were collected, crushed, and then dried at 100 °C for 48 h to remove moisture and mold. Then they were crushed and filtered to a size of 1 mm, transferred to Marinelli containers of 500 cm$^3$ each, tightly closed, and stored for at least 28 days to reach secular equilibrium between $^{222}$Rn and its decay products [29,30].

$^{226}$Ra, $^{232}$Th, and $^{40}$K activity concentrations were obtained by using a NaI (Tl) scintillation spectrometer Model 802 with a crystal size of 7.6 cm × 7.6 cm and a resolution of 7.5% at 661.6 keV, with a 1024-channels multichannel analyzer. It was calibrated in energy with reference sources containing $^{60}$Co (1173.23 and 1332.5 keV), $^{133}$Ba (383.9 keV), $^{54}$Mn (834.9 keV), $^{22}$Na (511 and 1274.5 keV), and $^{137}$Cs (661.6 keV) from the IAEA and in efficiency by a multi-energy standard analyzed under the same experimental conditions as the samples [31,32]. This standard is a blend of different radioactive sources, forming an energy range from 59.54 to 1836 keV [$^{60}$ C (1173.2 and 1332.5 keV), $^{137}$ Cs (661.6 keV), $^{152}$ Eu (1407.5, 1112, 964.079, and 778.9 keV), $^{40}$K (1460.8 keV), $^{137}$Cs (661.6 keV), $^{208}$Tl (2614.4 keV), and $^{228}$Ac (940.1 keV)].

After reaching secular equilibrium between $^{222}$Rn and its progeny, a gamma-ray line at 609.3 KeV of $^{214}$Bi was considered to determine the activity concentration of $^{226}$Ra, and a gamma-ray line at 969 KeV of $^{228}$Ac used to determine that of $^{232}$Th [33]. The spectra analysis was carried out by using GENIE 2000 (Canberra) software. $^{226}$Ra, $^{232}$Th, and $^{40}$K activity concentrations in soil were determined by the following equation [17,34,35]:

$$A = \frac{N_p}{t_c \times I_\gamma(E_\gamma) \times \varepsilon(E_\gamma) \times M} \tag{1}$$

where $N_P$ is the number of counts in a given peak area at energy, E; $\varepsilon(E_\gamma)$ the detection efficiency at energy, E; $t_c$ is the counting time of 100,000 s; $I_\gamma(E_\gamma)$ is the number of gamma-rays per decay of that nuclide at energy, E, and M, the mass in kg, of the sample. The uncertainty on the activity concentration ($\Delta A$) was obtained by the following equation [36,37]:

$$\frac{\Delta A}{A} = \sqrt{\left(\frac{\Delta N_p}{N_p}\right)^2 + \left(\frac{\Delta I_\gamma}{I_\gamma}\right)^2 + \left(\frac{\Delta \varepsilon}{\varepsilon}\right)^2 + \left(\frac{\Delta M}{M}\right)^2} \tag{2}$$

where $\Delta N$, $\Delta I_\gamma$, $\Delta \varepsilon$, and $\Delta M$ are the uncertainties in the count rate, emission probability found in the nuclear data tables, efficiency, and sample mass, respectively.

### 2.2.2. In Situ Radioactivity Measurements

They were performed simultaneously with sampling for laboratory analysis. Those measurements were made randomly at different points of the area with NucScout detector (portable Gamma Identifier-Quantifier—Dose Rate Meter) version 2018. It was installed one meter above the ground surface on a dry wooden stand. The measurement on a sample point took 45 min [38].

The NucScout is a high-sensitivity Na (Tl) gamma detector, with an integrated photo multiplier and high-voltage-supply cylindrical scintillation crystal, 2″ × 2″, with an energy range of 25 keV–3 MeV (optional from 10 keV to 1.6 MeV) and resolution < 8% (Cs-137/ 662 keV). It works with an integrated battery. The instrument has several options such as the selection of the measurement cycle, the reading, and the calculation of the results of a measurement. It has an integrated GPS that allows users to geolocate a sampling point, or even to bring out a Maps distribution of the different measured points. The data of the different measurements obtained on the site are stored on an a USB support or on an SD card and transferred for analysis and to a PC [39]. These data are visualized with the dvision software [40]. The detector is calibrated when connected to a PC, using the dconfig software [40]. $^{226}$Ra, $^{232}$Th, and $^{40}$K activity concentrations were obtained by using gamma lines at 609.3 KeV of $^{214}$Bi, 2614 KeV of $^{208}$Tl, and 1461 of $^{40}$k, respectively.

### 2.2.3. In Situ $^{222}$Rn in Soil Measurements

Measurements were taken at different locations with Markus 10 version 1.4. This instrument was developed by RADONOVA Laboratories to measure the volumic activity of $^{222}$Rn in soil, with about 3 kg and 16 keV of resolution energy (under vacuum); it is an ORTEC Ultra Silicon detector with dimensions of 220 × 122 × 80 cm$^3$, with a pumping capacity of 1.8 L/min every 30 s, under a limiting pressure of 0.96 bar. The duration of a measurement is typically 12 min, and its battery has a capacity of about 70 measurements before being fully recharged for 8 h [41].

The principle of measurement of the device consists of two steps. The first step is the pumping phase of the gas contained in soil. This is achieved with a probe buried one meter in the ground. The gas is sucked from the ground into the measuring chamber for a short period. The pumping phase is automatically stopped when the pressure in the probe drops; when the pressure rises, the pump starts again. The pumping phase is finally stopped when a capacity of 0.91 L is reached. The next one is automatically started and consists of the measurement. The measuring chamber is immediately switched on. An electric field pushes the radon progeny into the measuring chamber, where the alpha radiation they emit is recorded. These electric pulses recorded by the sensor are amplified and then filtered in the analysis channel, which allows only the counting of pulses corresponding to the energy coming from the $^{218}$Po. A latent measuring background is created in the ionization chamber of the system by filtering out the pulses from the $^{214}$Po. The evolution of the measurement can be read on a screen, with each hit recorded by the sensor, until the screen displays a fixed value to signify the end of the measurement.

*2.3. Radiological Hazards*

2.3.1. Ambient Equivalent Dose Rates and External Effective Dose

Ambient equivalent dose rates in air at distance of one meter on the ground surface are calculated using the conversion factor of 0.0417 $(nGy\ h^{-1})/(Bq\ kg^{-1})^{-1}$ for $^{40}K$, 0.462 $(nGy\ h^{-1})/(Bq\ kg^{-1})^{-1}$ for $^{226}Ra$, and 0.604 $(nGy\ h^{-1})/(Bq\ kg^{-1})^{-1}$ for $^{232}Th$ in the following equation [1].

$$D\left(nGy\ h^{-1}\right) = 0.462A_{Ra} + 0.604A_{Th} + 0.0417A_k \tag{3}$$

where $A_{Ra}$, $A_{Th}$, and $A_K$ are the mean concentrations of each radionuclide given in $(Bq\ kg^{-1})$. The effective dose due to external irradiation, E $(mSv\ y^{-1})$, was calculated by using the following formula [42,43]:

$$E\left(mSv\ y^{-1}\right) = F_c[F_{occ}F_b + (1 - F_{occ})] \times D \times T \times 10^{-6} \tag{4}$$

where $F_c = 0.7$ is the conversion coefficient of the absorbed dose in the air to effective dose received by adults, T is the exposure time expressed in hours, $F_b$ (0.98) is the impact factor of the building material experimentally obtained on the site, and $F_{occ} = 0.8$ is the occupancy coefficient [1].

2.3.2. External and Internal Hazard Index

External Hazard Index ($H_{ex}$)

The external hazard index was introduced to limit radiation exposure in the samples to a permissible dose-equivalent limit of 1.00 mSv $y^{-1}$ [1,9,44], and it is assessed by Equation (5):

$$H_{ext} = \frac{A_{Ra}}{370} + \frac{A_{Th}}{259} + \frac{A_K}{4810} \leq 1 \tag{5}$$

The external hazard index must not exceed the limit of unity for the radiological risk to be insignificant. The maximum value of $H_{ext}$ equal to unity corresponds to the upper limit of 370.00 Bq $kg^{-1}$ of $^{226}Ra$ [1,9,45].

Internal Hazard Index ($H_{in}$)

Furthermore, the deposition period of $^{222}Rn$ progeny in the pulmonary is also very dangerous [5,44]. In order to take this threat into account and reach the normal limit of 185 Bq $kg^{-1}$, the permissible value for $^{226}Ra$ is reduced by half to reach the limit of the unit. It is evaluated by using the following equation [44,46]:

$$H_{in} = \frac{A_{Ra}}{185} + \frac{A_{Th}}{259} + \frac{A_K}{4810} \leq 1 \tag{6}$$

*2.4. Excess Lifetime Cancer Risk (ELCR)*

The ELCR is the probability that an individual will contract or develop a radiation-induced cancer during his lifetime because of his exposure to ionizing radiation. It was estimated for this by using Equation (7) [9,13,47]:

$$ELCR = ELCR_{out} + ELCR_{in} \tag{7}$$

$ELCR_{out} = E_{out} \times D_L \times RF$ is the outdoor risk; $ELCR_{in} = E_{in}\ D_L \times RF$ is the indoor risk; $E_{out}$ and $E_{in}$ are the indoor and outdoor effective dose, respectively; DL is the average life expectancy of 70 years; and RF is the risk factor (risk of fatal cancer per mSv). In its publication 106, ICRP recommends value of $RF = 0.05 \times 10^{-3}$ $mSv^{-1}$ for induction to stochastic effects of members to the public [5].

*2.5. Excess Cancer Risk (ECR) Computer Using RESRAD-ONSITE and RESRAD-BUILD Codes*

Since most dwellings in the study are constructed with locally manufactured earthen or sand bricks, the $^{40}$K, $^{226}$Ra, and $^{232}$Th concentrations in soil are input data (contaminant on source parameters) at runtime by RESRAD-ONSITE and RESRAD-BUILD codes version 7.2 and 3.5, respectively.

RESRAD-ONSITE is used to assess the ECR due to these naturally occurring radionuclides in soil at the bauxite-bearing area of Fongo-Tongo. The site-specific characteristics of the area are listed in Table 1. The other parameters are used as defaults values [48]. Together, all the above parameters were considered in the evaluation of the risk factors.

**Table 1.** Input parameters for RESRAD codes.

| RESRAD-ONSITE | |
| --- | --- |
| Parameters | Site-Specific Data |
| Site-specific data | 25,000 m$^2$ |
| Cover depth | 1 m |
| Density of contaminated zone | 1.8 cm$^3$ g$^{-1}$ |
| Precipitation rate | 0.4473 m y$^{-1}$ |
| Wind speed | 1.2 m s$^{-1}$ |
| Well pump intake | 8 m |
| RESRAD-BULD | |
| Indoor/time fraction | 0.6 |
| Number of room/occupants | 1 |
| Deposition velocity | 0.01 m s$^{-1}$ |
| Resuspension rate | $5 \times 10^{-7}$ s$^{-1}$ |
| Room surface area and volume | 16 m$^2$ and 40 m$^3$ |
| Breathing rate | 18 m$^3$ d$^{-1}$ |
| Ingestion rate | 44,661 |
| Occupant location in the room | Centered |
| Shielding thickness | 0 |
| Type of source | Volume |
| Source geometry | Rectangular |
| Release air fraction | 0.1 |
| Radon diffusion rate | $2 \times 10^{-5}$ m s$^{-1}$ |
| Porosity | 0.1 |

RESRAD-BUILD allowed for the assessment of radiation doses received by a resident living or working in a house contaminated by radioactive materials. These doses are those from the different exposure pathways (external and internal, including inhalation of radon progeny inside the home). The radiological risk was estimated over the periods of 1, 10, 30, 50, 70, and 90 years of exposure. However, 85% of the dwellings in the area are made of mud bricks, usually produced on the same site, and samples of these earth bricks were analyzed to obtain the concentrations introduced as input data mentioned above. Table 1 presented the other input parameters.

*2.6. Radiation Hazard Index*

2.6.1. Gamma Radiation Hazard Index ($I_\gamma$)

The gamma radiation risk index was estimated from Equation (8) [47,49]:

$$I_\gamma = \frac{A_{Ra}}{300} + \frac{A_{Th}}{200} + \frac{A_K}{3000} \leq 1 \tag{8}$$

It is the index of nuclear energy level for external radiation due to specific activity of different natural radionuclides in a sample [50]. Its permissible limit is $I_\gamma = 1$ and corresponds to 0.3 mSv y$^{-1}$. It is used to evaluate the gamma-radiation risk level associated with naturally occurring radionuclides.

2.6.2. Alpha Radiation Hazard Index ($I_\alpha$)

The excess alpha radiation following radon inhalation from building materials is determined by using Equation (9) [51,52]:

$$I_\alpha = \frac{A_{Ra}}{200} \leq 1 \tag{9}$$

The upper limit of $I_\alpha$ is unity because a building material with a $^{226}$Ra concentration of less than 200 Bq kg$^{-1}$ cannot cause a minimum radon concentration greater than 200 Bq m$^{-3}$.

## 3. Results and Discussion

### 3.1. $^{226}$Ra, $^{232}$Th, and $^{40}$K Activity Concentrations

In Fongo-Tongo, the $^{226}$Ra, $^{232}$Th, and $^{40}$K activity concentrations obtained by laboratory and in situ methods ranged from 106 to 170 Bq kg$^{-1}$ and from 93 to 201 Bq kg$^{-1}$ for $^{226}$Ra; from 119 to 295 Bq kg$^{-1}$ and from 40 to 327 Bq kg$^{-1}$ for $^{232}$Th; and from 188 to 458 Bq kg$^{-1}$ and from 49 to 321 Bq kg$^{-1}$ for $^{40}$K.

In Dschang, the $^{226}$Ra, $^{232}$Th, and $^{40}$K activity concentrations range from 99 to 167 Bq kg$^{-1}$ and from 98 to 181 Bq kg$^{-1}$ for $^{226}$Ra, from 100 to 275 Bq kg$^{-1}$ and from 139 to 309 Bq kg$^{-1}$ for $^{232}$Th; and from 198 to 297 Bq kg$^{-1}$ and from 151 to 280 Bq kg$^{-1}$ for $^{40}$K. Figure 2 shows the box-plot distributions of these concentrations in laboratory (a) and in situ (b) for each locality and for the whole study area.

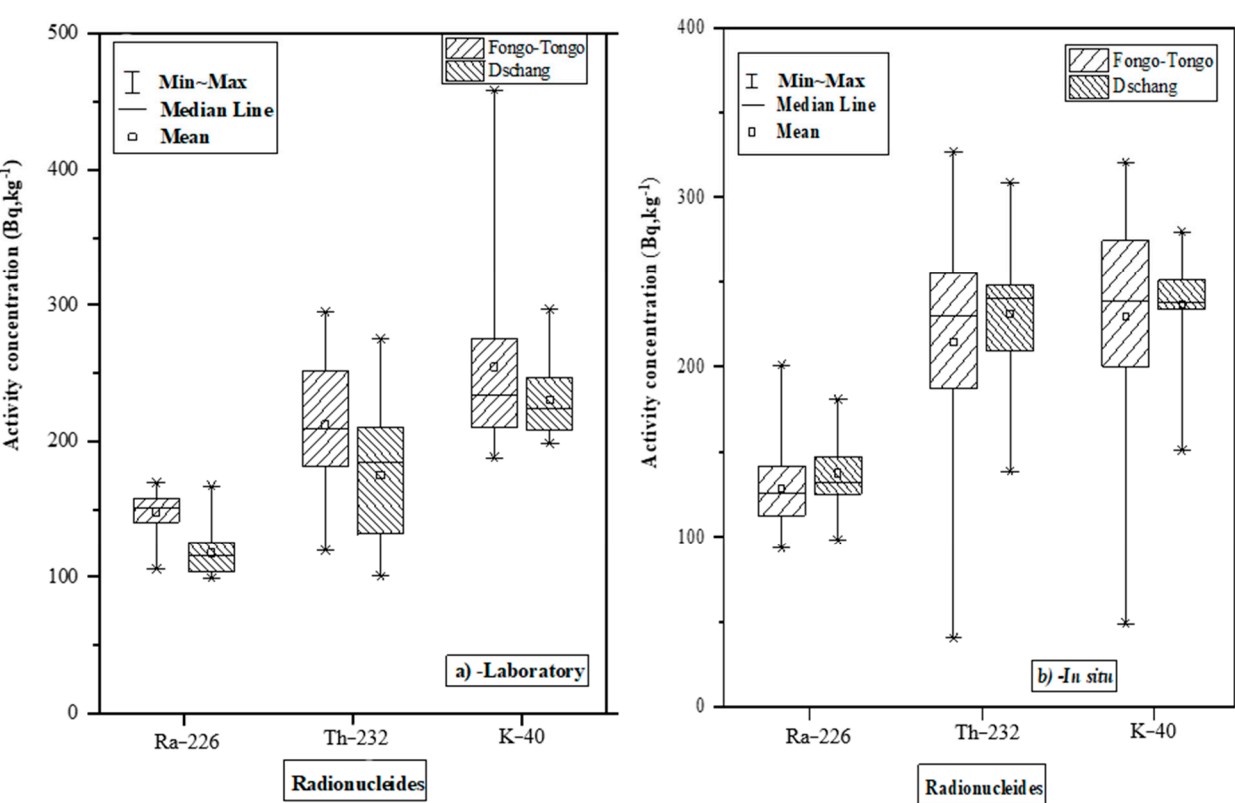

**Figure 2.** Boxplot distribution of activity concentration of $^{226}$Ra, $^{232}$Th, and $^{40}$K obtained by laboratory (**a**) and in situ (**b**) measurements.

According to Table 2, 50% of sampling points have a concentration higher than 151 Bq kg$^{-1}$, 209 Bq kg$^{-1}$, and 234 Bq kg$^{-1}$ for $^{226}$Ra, $^{232}$Th, and $^{40}$K, respectively, in laboratory measurements. Furthermore, the in situ measurements follow a lognormal distribution. Thus, the mean value is represented by the geometric mean, whereas laboratory measurements follow a normal distribution and are represented by the arithmetic mean.

**Table 2.** Statistical parameters of $^{226}$Ra, $^{232}$Th, $^{40}$K, and $^{222}$Rn concentrations obtained by in situ and laboratory measurements for the localities of Dschang and Fongo-Tongo.

| Locality | Parameters | Activity Concentration (Bq kg$^{-1}$) | | | | | | $^{222}$Rn (kBq m$^{-3}$) |
|---|---|---|---|---|---|---|---|---|
| | | Laboratory | | | In situ | | | |
| | | $^{226}$Ra | $^{232}$Th | $^{40}$K | $^{226}$Ra | $^{232}$Th | $^{40}$K | |
| Fongo-Tongo | Min–Max | 106–170 | 119–295 | 188–458 | 93–201 | 94–327 | 49–321 | 35–202 |
| | Median | 151 | 209 | 234 | 126 | 229 | 239 | 53 |
| | AM ± SD | 148 ± 23 | 212 ± 54 | 230 ± 28 | - | - | - | - |
| | GM(GSD) | - | - | - | 129 (16) | 214 (67) | 229 (54) | 69 (8) |
| Dschang | Min–Max | 99–167 | 100–275 | 198–297 | 98–181 | 139–309 | 151–280 | 48–255 |
| | Median | 116 | 185 | 224 | 132 | 240 | 238 | 62 |
| | AM ± SD | 118 ± 17 | 175 ± 46 | 230 ± 28 | - | - | - | - |
| | GM(GSD) | - | - | - | 138 (19) | 231 (35) | 237 (26) | 82 (14) |

AM, arithmetic mean; GM, geometric mean; SD, standard deviation; GSD, geometric standard deviation.

Soil samples analyzed in the laboratory have high concentrations of $^{226}$Ra and $^{232}$Th. As presented in Table 2, the minimum and maximum values of $^{226}$Ra obtained in laboratory and in situ measurements are, respectively, three and five times higher than the world average value of 35 Bq kg$^{-1}$ [1]. In the case of $^{232}$Th, they are two and four times higher than the world average value of 45 Bq kg$^{-1}$, respectively [1]. These high values of $^{226}$Ra and $^{232}$Th activity concentrations are also observed for the results obtained by in situ gamma spectrometry. The minimum values of $^{226}$Ra and $^{232}$Th are, respectively, three and two times higher than the world average value, while the maximum values are, respectively, six and seven times higher than the world average value [1]. Furthermore, the average values of $^{40}$K, as well as the maximum values for in situ and laboratory methods, are lower than 420 Bq kg$^{-1}$, the world average value [1].

Figure 1 shows that the investigated area extends over a geological structure covered by basaltic and trachytic granitic rocks [27,53]. The $^{226}$Ra, $^{232}$Th, and $^{40}$K activity concentrations differ from one point to another for the two techniques used: in situ and laboratory gamma spectrometry. This can be explained by the fact that radioactivity is not uniformly distributed in the soil [54]. It is reported that $^{238}$U, $^{232}$Th, and $^{40}$K have high concentrations in some rocks, such as syenite, granite, granulite, rhyolites, and plutonic [3,4,54]. The low concentrations of $^{40}$K can be explained by the phenomenon of leaching and transport of potassium elements to the surface due to the effects of erosion, drainage, and an accumulation of sediments in the seabed [55]. The transfer of ores by erosion or by eruptive voice can therefore considerably modify the content and concentrations of this radionuclide in the soil. It has low concentrations in basalt [3,4,54]. According to Figure 1, the presence of the above rocks can account for considerable variation in the concentrations of these primordial radionuclides from one site to another, as shown in Figure 2.

*3.2. In Situ $^{222}$Rn Concentration in Soil*

$^{222}$Rn concentrations at 1 m depth in soil, presented in Table 2, ranged from 35 to 202 kBq m$^{-3}$, with a mean value of 69 ± 40 kBq m$^{-3}$, in Fongo-Tongo; and from 48 to 255 kBq m$^{-3}$, with a mean value of 82 ± 56 kBq m$^{-3}$, in Dschang. According to Table 2, more than half of the sampled points have $^{222}$Rn in soil greater than or equal to 62 kBq m$^{-3}$ in Dschang and 53 kBq m$^{-3}$ in Fongo-Tongo. According to Figure 3, a majority of the radon concentrations in soil are above the value of 40 kBq m$^{-3}$, as represented by the red line. According to the Swedish risk assessment criteria, this latter value represents the limit for which a site presents a high radon-exposure risk [56]. The difference between $^{222}$Rn concentrations in soil from one location to another may be due to the geological structure and the mineralogical composition of the soil in the area [54,55]. The geological structure, the geochemical process of the soil, and the rate of gas emanation in the region are influenced by the permeability of the soil [27,28,49,53,54].

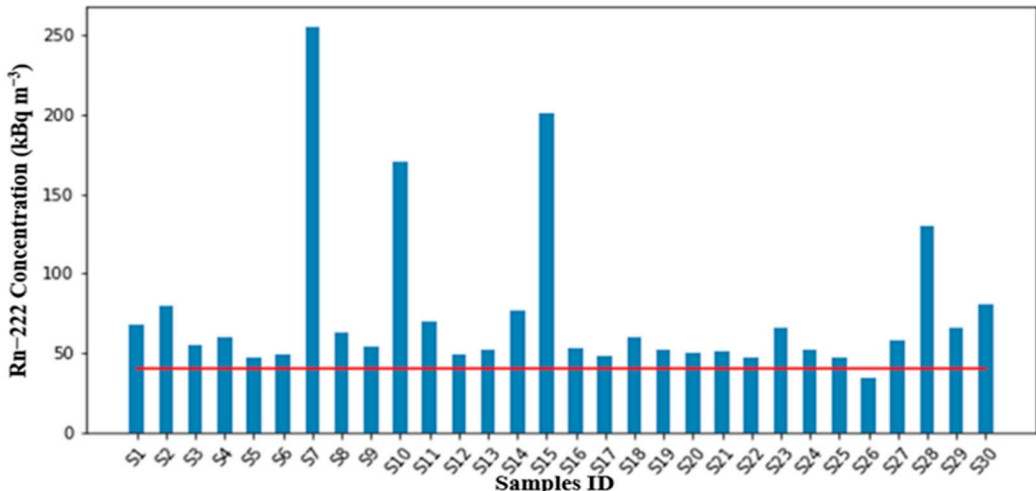

**Figure 3.** [222]Rn distribution in soil of the bauxite bearing area of Fongo-Tongo.

Table 2 shows that the average and maximum values of [222]Rn in soil in Dschang are higher than those in Fongo-Tongo. This is not the case with the [226]Ra values obtained in these two localities. This is probably due to the influence of soil moisture and porosity. In addition, the soil in the Fongo-Tongo may be more compact and moister than in Dschang [25,53]. In additional, Table 3 shows that activity concentrations of the primordial radionuclides in soil in Cameroon is higher than in some other regions of the world [51,52,57,58]. Nevertheless, [40]K concentrations are also high elsewhere than in the present study [46,52].

**Table 3.** Comparison of [226]Ra, [232]Th, [40]K, and [222]Rn activity concentration with other countries.

| Country | Activity Concentration (Bq kg$^{-1}$) | | | [222]Rn (kBq m$^{-3}$) | References |
|---|---|---|---|---|---|
| | [226]Ra | [232]Th | [40]K | | |
| Jordan | 57.7 ± 5.4 | 18.1 ± 1.4 | 138.1 ± 40.8 | | [46] |
| Egypt | 134.7 ± 24.1 | 131.8 ± 16.7 | 11,644 ± 550 | | [59] |
| India | 116.1 | 43.51 | 300.07 | - | [37] |
| Iraq | 58.44 | 19.38 | 321.76 | - | [9] |
| | 45.71 | 20.33 | 337.02 | | |
| Nigeria | 64.64 ± 28.10 | 110.18 ± 46.12 | 1190.10 ± 373.62 | | [51] |
| Australia | 38 | 45 | 635 | - | [10] |
| Germany | 84 | 72 | 463 | - | |
| Sweden | 75 | 94 | 734 | | |
| Japan | 38 ± 1 | 43 ± 1 | 590 | - | [8] |
| Cameroon | 14 ± 2 | 30 ± 3 | 103 ± 12 | 9 ± 2 | [60] |
| | - | 390 | 850 | - | [14] |
| | 124.9 | 157.3 | 670.9 | | [61] |
| | 166.18 | 170.04 | 94.54 | | [13] |
| | 118 ± 17 (138 ± 19) | 175 ± 46 (231 ± 35) | 230 ± 28 (237 ± 26) | 82 ± 56 | Present study |
| | 148 ± 23 (129 ± 16) | 212 ± 54 (214 ± 67) | 230 ± 28 (229 ± 54) | 69 ± 40 | |

### 3.3. Correlation between [222]Rn and [226]Ra in Soil

According to Figure 4, it is shown that [222]Rn concentrations in soil are directly related to those of [226]Ra measured at the site and in soil samples collected in the area. The R$^2$ = 0.88 and R$^2$ = 0.86 values were found between [222]Rn and [226]Ra concentrations for the laboratory method and in situ method, respectively. These high values of the coefficients obtained for each case reveal that [222]Rn and [226]Ra are strongly correlated. Similarly, the Pearson correlation coefficient determined for both sets of measurements is equal to 0.92 for the

laboratory and 0.90 for in situ. These respective Pearson correlation coefficients for the two series confirm the strong correlation between the two radionuclides.

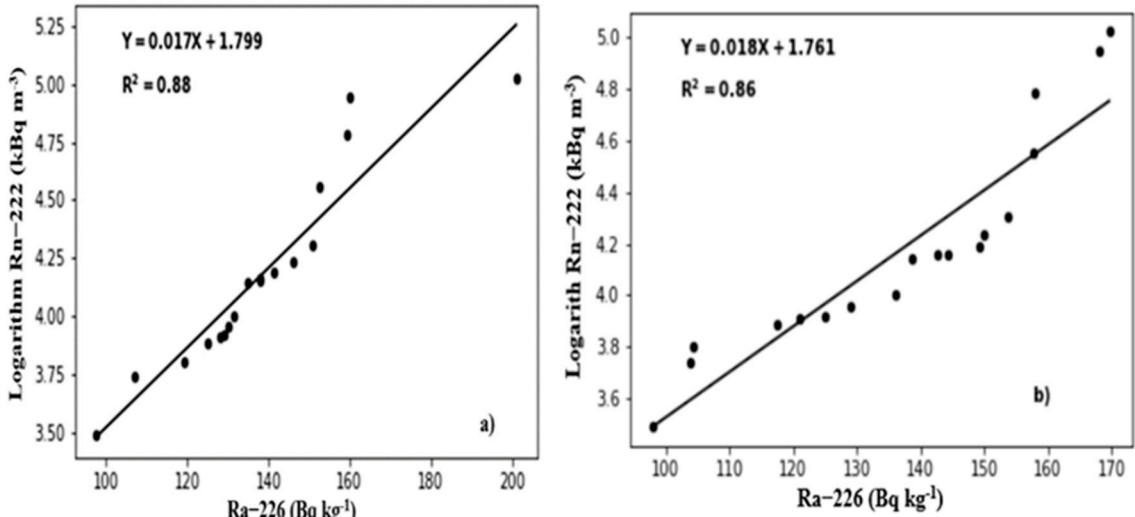

**Figure 4.** Correlation between $^{222}$Rn/$^{226}$Ra concentrations in soil: (**a**) laboratory gamma-ray spectrometry and (**b**) in situ gamma-ray spectrometry.

For the values observed between 110 and 150 Bq kg$^{-1}$ (Figure 4a) and values between 130 and 160 Bq kg−1 (Figure 4b), the residual is relatively constant, and for the extreme values, it increases slightly, which shows a dispersion of the maximum values from the median value. This can be justified by the fact that the number of samples of the different datasets is not very high to make the scatterplot dense enough and have a better regression. That is, the closer in the series values are to each other, the better correlation the coefficient and the stronger correlation intensity. This also means that, the smaller the standard deviation is between the data, the better the regression and the stronger the correlation between the two radionuclides.

The high values of $^{222}$Rn concentration in soil gas at some locations certainly originate from the deep sources of permeable soil, which allows $^{222}$Rn to easily escape from its cradle, which is $^{226}$Ra, and migrate to the free surface of the soil. In other words, the high emanation of $^{222}$Rn at a measurement point is closely related to the nature of underlying rock, geochemical process, physicochemical soil properties, and $^{226}$Ra content in soil. The correlation observed between these concentrations depends on the geological structure of the area [62]. Similar results are reported in previous studies [47,63].

As shown in Figure 1, the area contains different rock formations, such as granite, basalt, gneiss, and trachyte. In addition, it is characterized by a deposit of bauxite ores [26,27]. Granite, mined in quarries in Dschang and Fongo-Tongo, is probably a potential source of $^{226}$Ra distributed in the area. It is known to have a high content of uranium, thorium, and potassium at high temperatures in these rocks [54]. $^{222}$Rn emanation may therefore be stronger in an area underlain by granitic bedrock.

Figure 5 shows the distribution map of $^{222}$Rn and $^{226}$Ra activity concentrations in the soil of the study area. It shows that the activity concentration of $^{226}$Ra in soil increases with the $^{222}$Rn concentration in its close proximity.

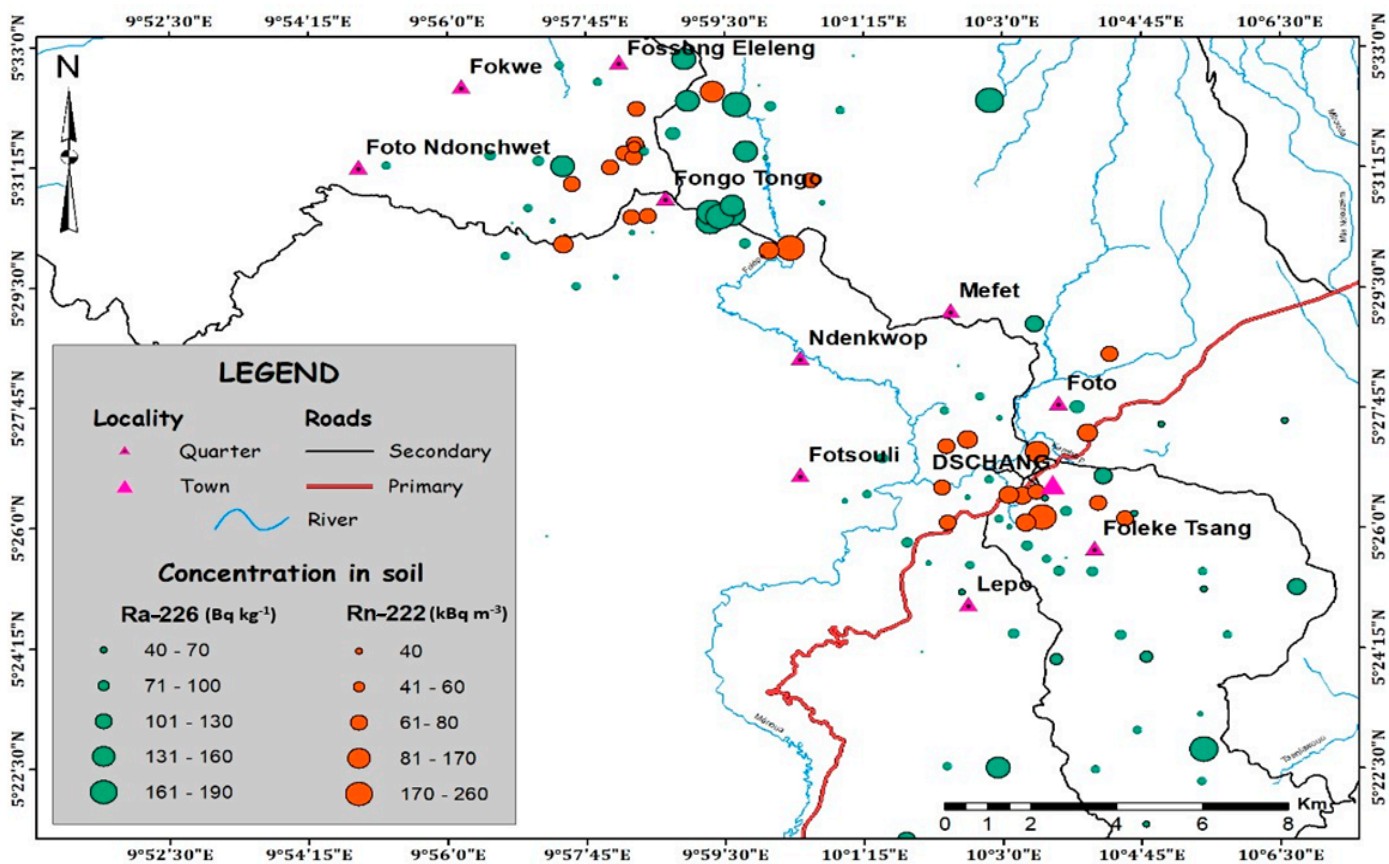

**Figure 5.** Map distribution of $^{222}$Rn and $^{226}$Ra concentrations in the soil of the study area.

*3.4. Radiological Hazards*

3.4.1. Ambien Equivalent Dose Rate (AEDR) and Annual External Effective Dose (AEED)

The AEED obtained in the laboratory ranged from 0.58 to 1.62 mSv y$^{-1}$, with a mean value of 1.27 ± 0.27 mSv y$^{-1}$, in Fongo-Tongo; and from 0.73 to 1.46 mSv y$^{-1}$, with a mean value of 1.05 ± 0.17 mSv y$^{-1}$, in the Dschang locality. According to Table 4 the average values for the whole study area are above the safety limit of 1.00 mSv y$^{-1}$ [1].

**Table 4.** Summary of the different radiological parameters obtained in laboratory.

| Locality Parameters | Fongo-Tongo | | | | | Dschang | | | | | Limit |
|---|---|---|---|---|---|---|---|---|---|---|---|
| | Min | Max | Med | AM | SD | Min | Max | Med | AM | SD | |
| AEDR (nGy/y) | 130 | 265 | 211 | 207 | 37 | 119 | 238 | 172 | 170 | 31 | 1 |
| AEED (mSv) | 0.8 | 1.62 | 1.29 | 1.27 | 0.22 | 0.73 | 1.46 | 1.05 | 1.04 | 0.19 | 1 |
| $H_{in}$ | 1.36 | 2.81 | 1.82 | 1.88 | 0.37 | 1.36 | 1.92 | 1.57 | 1.6 | 0.15 | 1 |
| $H_{out}$ | 1.07 | 2.35 | 1.41 | 1.48 | 0.33 | 1.29 | 1.58 | 1.29 | 1.29 | 1.02 | 1 |
| $ELCR_{in}$ | 1.68 | 3.4 | 2.71 | 2.67 | 0.47 | 1.53 | 3.06 | 2.21 | 2.18 | 0.4 | |
| $ELCR_{out}$ | 1.12 | 2.26 | 1.81 | 1.78 | 0.31 | 1.02 | 2.04 | 1.47 | 1.46 | 0.27 | |
| ELCR | 2.59 | 5.66 | 4.52 | 4.44 | 0.78 | 2.55 | 5.11 | 3.69 | 3.64 | 0.66 | |
| $I_\alpha$ | 0.53 | 0.85 | 0.76 | 0.74 | 0.08 | 0.49 | 0.84 | 0.58 | 0.59 | 0.09 | 1 |
| $I_\gamma$ | 1.02 | 2.11 | 1.66 | 1.64 | 0.3 | 0.93 | 1.9 | 1.36 | 1.34 | 0.25 | 1 |

AM, arithmetic mean; GM, geometric mean; SD, standard deviation; GSD, geometric standard deviation.

According to Table 5, the AEDR at one meter above ground surface ranged from 130 to 265 nGy h$^{-1}$ and from 119 to 238 nGy h$^{-1}$ at Fongo-Tongo and Dschang, respectively, with an average of 207 ± 37 nGy h$^{-1}$ and 170 ± 31 nGy h$^{-1}$ for soil samples analyzed in the laboratory. It ranged from 95 to 264 nGy h$^{-1}$ and from 69 to 126 nGy h$^{-1}$, with a mean value of 198 ± 45 nGy h$^{-1}$ and 96 ± 14 nGy h$^{-1}$, for the in situ measurement in

Fongo-Tongo and Dschang, respectively. The mean values of the current studies are all above the value set of 60 nGy h$^{-1}$ [1].

**Table 5.** Summary of the different radiological parameters obtained by in situ.

| Locality | Fongo-Tongo | | | | | Dschang | | | | | Limit |
|---|---|---|---|---|---|---|---|---|---|---|---|
| Parameters | Min | Max | Med | AM | SD | Min | Max | Med | AM | SD | |
| AEDR (nGy/y) | 95 | 264 | 210 | 198 | 45 | 69 | 126 | 94 | 96 | 14 | 1 |
| AEED (mSv) | 0.58 | 1.62 | 1.27 | 1.22 | 0.28 | 0.42 | 0.77 | 0.58 | 0.59 | 0.08 | 1 |
| H$_{in}$ | 0.9 | 2.01 | 1.65 | 1.57 | 0.32 | 1.25 | 2.04 | 1.68 | 1.68 | 0.19 | 1 |
| H$_{out}$ | 0.56 | 1.64 | 1.3 | 1.22 | 0.29 | 0.92 | 1.64 | 1.32 | 1.31 | 0.16 | 1 |
| ELCR$_{in}$ | 1.22 | 3.4 | 2.71 | 2.56 | 0.58 | 0.89 | 1.62 | 1.21 | 1.24 | 0.17 | |
| ELCR$_{out}$ | 0.81 | 2.7 | 1.8 | 1.7 | 0.31 | 0.59 | 1.08 | 0.81 | 0.83 | 0.11 | |
| ELCR | 2.03 | 5.67 | 4.51 | 4.26 | 0.97 | 1.48 | 2.7 | 2.01 | 2.07 | 0.28 | |
| I$_\alpha$ | 0.47 | 1.01 | 0.63 | 0.64 | 0.11 | 0.49 | 0.9 | 0.66 | 0.69 | 0.09 | 1 |
| I$_\gamma$ | 0.72 | 2.12 | 1.67 | 1.58 | 0.37 | 1.19 | 2.12 | 1.7 | 1.69 | 0.21 | 1 |

AM, arithmetic mean; GM, geometric mean; SD, standard deviation; GSD, geometric standard deviation.

### 3.4.2. External and Internal Radiation Hazard Index

External Hazard Index

The obtained values of H$_{ext}$ are presented in Table 4. The average values are 1.48 at Fongo-Tongo and 1.32 at Dschang. H$_{ext}$ values are greater than unity, and therefore, it can be recommended to the populations of those sites to use earth as a building construction material, except in some places where the level of natural radioactivity is relatively high.

Internal Hazard Index

The statistical parameters from H$_{in}$ are summarized in Table 4. The maximum values of H$_{in}$ are 2.81 and 2.04, with an average value of 1.88 and 1.68, in Fongo-Tongo and Dschang, respectively. H$_{in}$ values are also greater than unity [64]. Nevertheless, to avoid excessive internal exposure to $^{222}$Rn in these localities, the use of earth can be recommended as a building material, provided that there is good ventilation and air circulation in the rooms of the dwelling.

### 3.4.3. Excess Lifetime Cancer Risk (ELCR)

The ELCR statistical parameters' values obtained by gamma spectrometry in laboratory and in situ are summarized in Table 4. They ranged from $2.03 \times 10^{-3}$ to $5.67 \times 10^{-3}$, with a mean value of $4.44 \times 10^{-3}$, in Fongo-Tongo; and from $1.48 \times 10^{-3}$ to $5 \times 11\ 10^{-3}$, with a mean value of $3.64 \times 10^{-3}$, in Dschang. The mean values of ECR in Fongo-Tongo and Dschang were, respectively, 1.29 and 1.06 times higher than $0.29 \times 10^{-3}$, the UNSCEAR recommended limit value [1]. However, the risk values obtained could be overestimated if, in addition to the above risk, the risk due to radioactivity from building materials was taken into account, because more than 70% of the houses in the area use mainly mud bricks as building material.

### 3.5. Long-Term ECR Analysis Using RESRAD-ONSITE and RESRAD-BUILD Computer Codes

As shown in Figure 6, the total ECR calculated with RESRAD-ONSITE decreased progressively over the years, from the maximum value of $8.58 \times 10^{-3}$ obtained at the dates T = 1 and T = 1 year to the value of $7.41 \times 10^{-3}$ obtained at T = 100 years before decreasing significantly. This remarkable decreasing may be due to the self-absorption of building materials or to the process of radioactive decay [65].

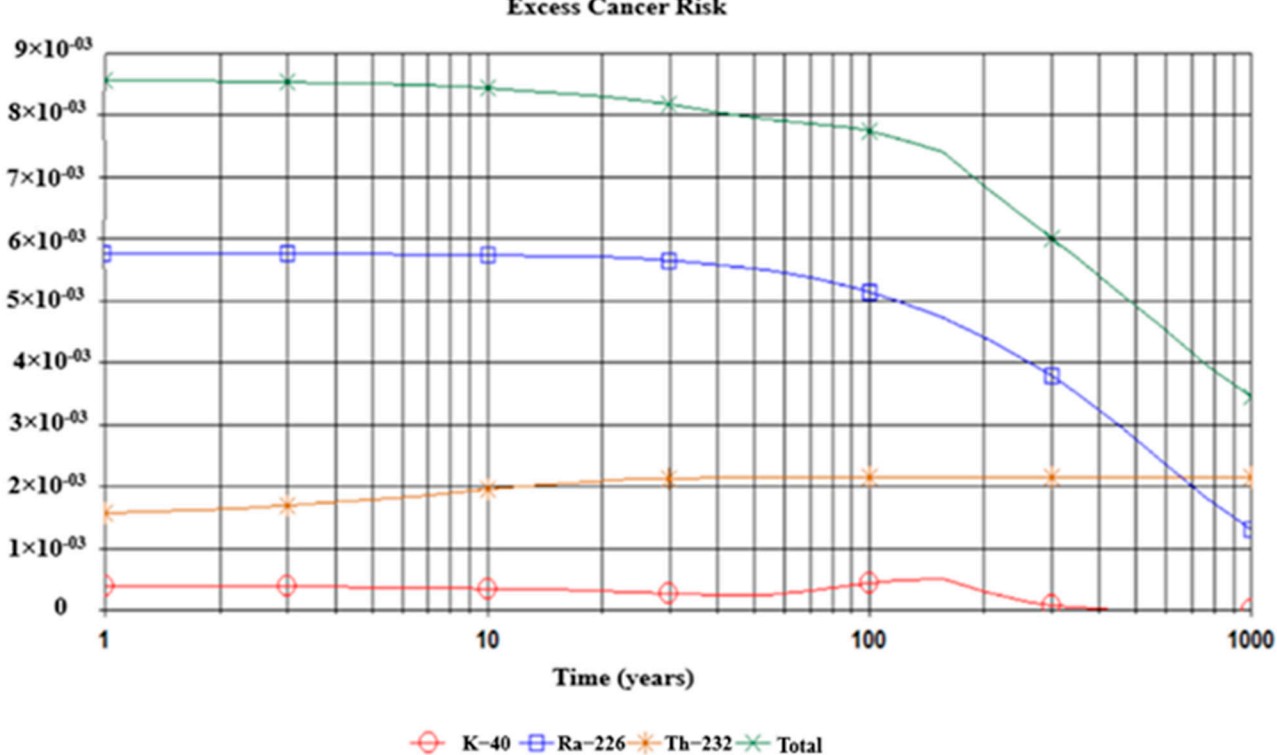

**Figure 6.** Long-term plotting of ECR for all exposure pathways and for each primordial radionuclide.

Similarly, $^{226}$Ra is the major contributor to the total ECR at about 70% in the first year. This contribution decreases slightly over the years before dropping significantly after 100 years. The maximum value of risk due to $^{226}$Ra obtained at T = 10 years is $7.372 \times 10^{-6}$. The ECR due to $^{232}$Th, on the other hand, is inversely proportional to that of $^{226}$Ra over the period from 1 to 40 years, where it becomes practically constant, and the maximum value obtained at T = 50 years is $9.250 \times 10^{-6}$. As for $^{40}$K, its contribution to the total risk remains the smallest, but it shows some slight variations before decreasing to zero. Similar results were observed in studies conducted in the cobalt–nickel region of Lomié in Eastern Cameroon [66]. Table 5 summarizes the total ECR for initially existent radionuclides and pathways at T = 0, 1, 10, 30, 50, and 100 years.

RESRAD-BUILD assessed the total risk due to radioactivity from soil used in the manufacture of bricks as a building material. The results obtained for the different exposure routes and for each nuclide as a function of time are summarized in Table 2. The maximum value of the total excess risk obtained at T = 30 years is $5.19 \times 10^{-2}$ for all the summed routes. Similarly, the value of the total excess risk for all summed nuclides obtained at T = 30 years is $1.89 \times 10^{-2}$. However, it should be noted that the external pathway is the one that contributes the most to the total excess risk. The maximum risk value for this pathway, which is $2.33 \times 10^{-2}$, was obtained at T = 30 years. Nevertheless, the decrease observed beyond 30 years for the external route would be due to the self-absorption of building materials [15,67,68]. Similar results were obtained in the work carried out in the Poli uranium region [17], in the bauxite zones of Southern Adamawa [16], and in some localities of the Centre Region, Cameroon [67].

The results presented in Table 6 show that $^{226}$Ra is the main contributor to the total excess risk compared to $^{232}$Th and $^{40}$K. The risk due to $^{226}$Ra increases progressively with time until reaching an increasing threshold after 70 years. The occurrence of this radionuclide in high concentrations in building materials increases the probability of accumulation of high indoor radon concentration [68]. Figure 7 represents the long-term total ECR for each radionuclide.

**Table 6.** Total ECR for initially existent radionuclides and pathways and fraction of total risk.

| T (Years) | Ground | Inhalation | Radon | Plant | Meat | Milk | Soil | Total |
|---|---|---|---|---|---|---|---|---|
| 0 | $1.74 \times 10^{-3}$ | $6.32 \times 10^{-6}$ | $4.89 \times 10^{-3}$ | $1.56 \times 10^{-3}$ | $2.24 \times 10^{-4}$ | $1.44 \times 10^{-4}$ | $2.22 \times 10^{-5}$ | $8.58 \times 10^{-3}$ |
| 1 | $1.74 \times 10^{-3}$ | $6.32 \times 10^{-6}$ | $4.89 \times 10^{-3}$ | $1.56 \times 10^{-3}$ | $2.24 \times 0^{-4}$ | $1.44 \times 10^{-4}$ | $2.22 \times 10^{-5}$ | $8.58 \times 10^{-3}$ |
| 3 | $1.73 \times 10^{-3}$ | $6.32 \times 10^{-6}$ | $4.87 \times 10^{-3}$ | $1.55 \times 10^{-3}$ | $2.18 \times 10^{-4}$ | $1.42 \times 10^{-4}$ | $2.22 \times 10^{-5}$ | $8.54 \times 10^{-3}$ |
| 10 | $1.72 \times 10^{-3}$ | $6.30 \times 10^{-6}$ | $4.81 \times 10^{-3}$ | $1.53 \times 10^{-3}$ | $2.06 \times 10^{-4}$ | $1.36 \times 10^{-4}$ | $2.20 \times 10^{-5}$ | $8.43 \times 10^{-3}$ |
| 30 | $1.69 \times 10^{-3}$ | $6.27 \times 10^{-6}$ | $4.67 \times 10^{-3}$ | $1.49 \times 10^{-3}$ | $1.76 \times 10^{-4}$ | $1.24 \times 10^{-4}$ | $2.17 \times 10^{-5}$ | $8.17 \times 10^{-3}$ |
| 100 | $1.60 \times 10^{-3}$ | $6.18 \times 10^{-6}$ | $4.19 \times 10^{-3}$ | $1.38 \times 10^{-3}$ | $1.16 \times 10^{-4}$ | $9.75 \times 10^{-5}$ | $2.04 \times 10^{-5}$ | $7.41 \times 10^{-3}$ |

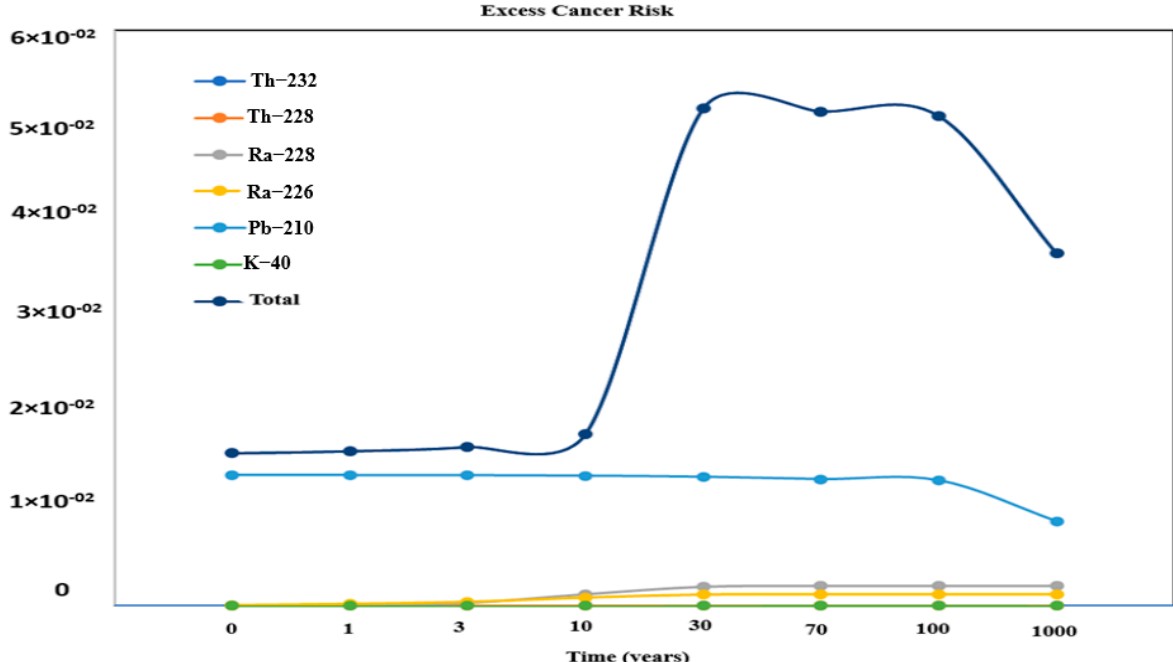

**Figure 7.** Long-term total excess risk for each nuclide.

According to Table 7, the pathway that contributes most to the total cancer risk is the external pathway. Like the other pathways, the risk increases until it reaches a value of $2.33 \times 10^{-2}$ at T = 30 years. similarly, the total cancer risk also increases and reaches a value of $5.19 \times 10^{-2}$ at the same date.

**Table 7.** Total risk of excess cancer for all exposure pathways.

| Pathway Detail of Risks | ELCR T (Years) | | | | | | |
|---|---|---|---|---|---|---|---|
| | T = 0 | T = 1 | T = 3 | T = 10 | T = 30 | T = 70 | T = 100 |
| External | $1.57 \times 10^{-2}$ | $1.59 \times 10^{-2}$ | $1.63 \times 10^{-2}$ | $1.76 \times 10^{-2}$ | $2.33 \times 10^{-2}$ | $2.32 \times 10^{-2}$ | $2.30 \times 10^{-2}$ |
| Deposition | $5.15 \times 10^{-9}$ | $5.22 \times 10^{-9}$ | $5.31 \times 10^{-9}$ | $5.64 \times 10^{-9}$ | $3.68 \times 10^{-3}$ | $3.66 \times 10^{-3}$ | $3.63 \times 10^{-3}$ |
| Immersion | $4.50 \times 10^{-11}$ | $4.54 \times 10^{-11}$ | $4.65 \times 10^{-11}$ | $5.06 \times 10^{-10}$ | $3.68 \times 10^{-3}$ | $3.66 \times 10^{-3}$ | $3.63 \times 10^{-3}$ |
| Inhalation | $1.18 \times 10^{-6}$ | $1.23 \times 10^{-6}$ | $1.36 \times 10^{-6}$ | $1.76 \times 10^{-6}$ | $3.01 \times 10^{-3}$ | $2.99 \times 10^{-3}$ | $2.96 \times 10^{-3}$ |
| Radon | $2.20 \times 10^{-4}$ | $2.27 \times 10^{-4}$ | $2.49 \times 10^{-4}$ | $3.30 \times 10^{-4}$ | $4.08 \times 10^{-3}$ | $4.06 \times 10^{-3}$ | $4.03 \times 10^{-3}$ |
| Ingestion | $5.39 \times 10^{-8}$ | $5.88 \times 10^{-8}$ | $6.82 \times 10^{-8}$ | $8.39 \times 10^{-8}$ | $1.89 \times 10^{-2}$ | $1.88 \times 10^{-2}$ | $1.86 \times 10^{-2}$ |
| Total | $1.59 \times 10^{-2}$ | $1.61 \times 10^{-2}$ | $1.66 \times 10^{-2}$ | $1.79 \times 10^{-2}$ | $5.19 \times 10^{-2}$ | $5.15 \times 10^{-2}$ | $5.11 \times 10^{-2}$ |

### 3.6. Radiation Hazard Index

#### 3.6.1. Gamma Radiation Hazard Index, $I_\gamma$

The results obtained give maximum values of $I_\gamma$ equal to 2.12 and 2.67 at Fongo-Tongo and equal to 2.12 and 1.90 at Dschang for in situ and laboratory measurements, respectively,

which are significantly greater than or equal to 2 to 2.7 times the maximum permissible value [50]. Similarly, the mean values of 1.69 and 1.34 at Dschang and 1.58 and 1.64 at Fongo-Tongo are also above the recommended limit. Thus, the land in the region could be exempted from all types of restrictions with respect to radiological risks, except at certain locations where $I_\gamma$ is very high.

### 3.6.2. Alpha Radiation Hazard Index, $I_\alpha$

The average values of $I_\alpha$ are reported in Table 4 and are below the reference limit value of unity for both study sites. Therefore, the soil bricks made at the study sites can be used as a building material in these two localities without exposing the inhabitant to a major risk of induction of lung cancer, because the $I_\alpha$ is below the safety limit recommended by UNSCEAR.

## 4. Conclusions

The current work was performed to study the $^{222}$Rn and $^{226}$Ra correlation that may exist in soil and assess the onsite and in-dwellings long-term ECR in the bauxite-bearing area of Fongo-Tongo. To achieve this, gamma spectrometry by in situ and laboratory was used to determine activity concentrations of $^{226}$Ra in soil. A strong correlation was found between $^{226}$Ra determined from the two methods and $^{222}$Rn in the soil. The $^{222}$Rn measurement in soil is therefore an excellent predictor of $^{226}$Ra and vice versa. Radiological parameters such as AEED, $H_{in}$, $H_{ext}$, ELCR, $I_\gamma$, and $I_\alpha$ were also determined to assess the level of radiological exposure of the public. Their values were all higher than the various corresponding safety limits recommended by UNSCEAR. The cancer risk assessed with RESRAD-ONSITE following exposure to the various radionuclides decreases from the first to the hundredth year for all the primordial radionuclides. The risk tends toward zero after the thousandth year. The maximum value of the total cancer risk of $8.58 \times 10^{-3}$ was observed at t = 1 year. It should also be noted that the contribution of $^{226}$Ra to cancer risk is high compared to that of $^{232}$Th. $^{226}$Ra is therefore the major contributor to cancer risk. A decrease in the contribution of all exposure pathways is observed from t = 1 year to t = 100 years. The risk tends to decrease considerably after 100 years. The cancer risk due to inhalation of radon and its progeny increases and reaches a peak of $3.01 \times 10^{-3}$ at t = 70 years. It should be noted that RESRAD-BUILD evaluates the risk related to radon and thoron according to the concentration of radium and thorium. Given the high concentrations of $^{232}$Th in soil samples from the current study area, the contribution of thoron ($^{220}$Rn) to cancer risk is high. Nevertheless, the observed decrease over time for all pathways and all radionuclides could be due to the self-absorption of building materials.

**Author Contributions:** Conceptualization, L.B.D. and S.; methodology, L.B.D., G.S.B. and S.; software, L.B.D., O.B.M. and J.E.N.N.II validation, L.B.D., G.S.B., and S.; formal analysis, L.B.D., J.E.N.N.II and S.; writing—original draft preparation, L.B.D., O.B.M. and G.S.B.; funding acquisition, S. All authors have read and agreed to the published version of the manuscript.

**Funding:** This work was partly supported by the International Atomic Energy Agency (IAEA) within the framework of the Technical Cooperation (TC) Project CMR9009. The Government of Cameroon supported field works through the Public Investment Budget 2020 and 2021 of the Ministry of Scientific Research and Innovation.

**Institutional Review Board Statement:** Not applicable.

**Informed Consent Statement:** Not applicable.

**Data Availability Statement:** The authors confirm that the data supporting the findings of this study are available within the article.

**Acknowledgments:** The Ministry of Scientific Research and Innovation of Cameroon is acknowledged for funding the field works through the Public Investment Budget 2021 allocated to the Institute of Geological and Mining Research.

**Conflicts of Interest:** The authors declare no conflict of interest.

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
