# Peer review of "Correlation between Ground 222Rn and 226Ra and Long-Term Risk Assessment at the at the Bauxite Bearing Area of Fongo-Tongo, Western Cameroon"

_radiation, doi:10.3390/radiation2040029_

Round 1

Reviewer 1 Report

This manuscript is about the natural radioactivity at the bauxite-bearing area of Fongo-Tongo, western-Cameroon. 

It is a well-written study with a logical structure. I have only some small comments or questions. 

1. Type and manufacturer of the different devices (Line 95 - NaI(Tl), Line 106 - GENIE software, Line 117- NucScout detector)

2. What was the activity of the reference sources, which was used for the efficiency determination?

3. Line 108-111: missing units (where is necessary)

4. Eq. 4 - what does T mean in the equation?

5. Line 178 and 182 you used Bq.kg-1, but before you did not use the point. Please unify it.

6. Table 1. improving the writing of the units.

7. Table 2. Is the difference between the laboratory and in-situ measurement significant? Please justify the answer by statistical analysis.

8. Fig 3. What does the red line mean on the figure? 

9. Fig 5. In the region of Dschang, I see big red circles (high radon conc.) and small green circles (low Ra conc.). It is not consistent with fig 4. Please, explain this conflict.

10. Table 4. is difficult to understand, the values are too close to each other. Is it possible to split the table?

Author Response

COMMENT

  1. Type and manufacturer of the different devices (Line 95 - NaI(Tl), Line 106 - GENIE software, Line 117- NucScout detector).

RESPONSE

NaI(Tl) model (802) and GENIE-2000 software (Canberra). These details have been added in the revised manuscript. (Line 97 and Line 112).

COMMENT

2          What was the activity of the reference sources, which was used for the efficiency determination?

RESPONSE

The activities of the reference sources used for the calibration of the spectrometer have been inserted in the revised manuscript Line 106-108.

COMMENT

3          Line 108-111: missing units (where is necessary)

RESPONSE

The units have been inserted in the revised manuscript.

COMMENT

  • 4 - what does T mean in the equation?

RESPONSE

tc is the counting time of 100, 000s. It has been added in the revised manuscript.

COMMENT

  • Line 178 and 182 you used Bq.kg-1, but before you did not use the point. Please unify it.

RESPONSE

The units have been unified in the revised manuscript.

COMMENT

6    Table 1. improving the writing of the units.

RESPONSE

Thank you for this observation.  The correction has been made in the revised manuscript.

COMMENT

7  Table 2. Is the difference between the laboratory and in-situ measurement significant? Please justify the answer by statistical analysis.

RESPONSE

Indeed, the aim of this work was not to make a comparative study between in situ and laboratory measurements. It is very difficult to compare with a good approximation the results of these two methods. The in situ measurements simply allow us to have a general idea on the veracity of the results of the laboratory measurements. Nevertheless, the results obtained from the two techniques as a whole reveal many similarities at certain points where the two techniques for measuring natural radioactivity were carried out.

Some of the differences observed in the results can be explained by the fact that in-situ γ-spectrometry gives a representation of the source concentration on a wide horizontal plane, up to 50 m in radius and 10 cm in depth. In contrast, laboratory γ-spectrometry measures radioactivity in a soil sample collected over an area of 1 m 2 (approximately 70 cm radius). Moreover, the matrix sampled (the soil) is not the same in composition in the two processes: in in-situ γ spectrometry, the sampled soil is generally compact and inhomogeneous, more or less humid, with various rocks, vegetation and debris of all kinds (plants, minerals, etc.). On the other hand, in laboratory γ-spectrometry, the soil matrix is dried, homogeneous, free of any debris and not compact. In addition, it is sieved so that the solid particles that constitute it have approximately the same volume. It is a matrix in which the radioelements it contains have undergone disturbances during sampling and must reach secular equilibrium in order to be able to reveal with the greatest precision, their different activities. If we take into account the uncertainties related to each measurement technique (in-situ, laboratory), the values of the results obtained are approximately the same.

COMMENT

8  Fig 3. What does the red line mean on the figure? 

RESPONSE

The red line in Figure 3 corresponds to the value of the radon concentration in soil for which the risk is considered high according to the Swedish risk assessment criteria. This clarification has been made in the revised manuscript. Lines 407-409

COMMENT

9  Fig 5. In the region of Dschang, I see big red circles (high radon conc.) and small green circles (low Ra conc.). It is not consistent with fig 4. Please, explain this conflict.

RESPONSE

Indeed, the values of the concentrations in Figure 4 are logarithmic. By applying the logarithmic function on the value of the radon concentrations in the soil observed in Figure 5, the result obtained would be in agreement with those of Figure 4.

COMMENT

10  Table 4. is difficult to understand, the values are too close to each other. Is it possible to split the table?

RESPONSE

It is true, this table is very cluttered.  It has been divided into two parts in the revised manuscript. The in situ results have been separated (Table 5), from the laboratory results (Table 4).

Reviewer 2 Report

It is an interesting article on the correlation between Ra226 and Rn222 in soil and its long-term risk in an area of Cameroon.

As a review, it should be taken into account the following points:

Abstract

Lines 27-28: Values expressed are mean ± standard deviation or uncertainty? Please explain.

Introduction

Line 63: please replace thickness with depth.

Materials and Methods

Line 92: Please check the size: 1 µm or 1 mm?

Line 102: Check superscript (137Cs).

Line 105: Check superscript (228Ac) and energy (usually 911 keV but 969 keV could be correct).

Lines 108-110: Express units to counting time, declare type of counts.

Eq. 2: Please check carefully: shouldn't the addends be squared?

Lines 122-128: consider writing in a more technical, less “commercial” way.

2.2.3. Please explain how or why Markus 10 doesn't wait until secular equilibrium.

Lines 157-159: Conversion factors in (nGyh-1/Bqkg-1), the split is missing.

Check Eq.4 everything correct in square brackets?

Line 171: 2.3.1 should be 2.3.3

Lines 172 and 179: index better than indexes

Eq. 9: Please check if AK/300 or AK/3000

Lines 227-240: summarize this information because Table 2 already shows all the data.

Line 281: 68 or 53 kBqm-3.

Figure 3. Missing units for Rn222.

Line 307: please, check if the sentence is finished.

Lines 332-334: Please, review the text.

Figure 5. Units are missing.

Line 338: Check ambient spelling.

Lines 343-387: please carefully check all numerical values and units both in the text and in table 4.

Line 375: consider replace earth with soil.

Figure 6. Consider if it makes sense to represent up to 1000 years.

Table 6 and Figure 7 could be repetitive.

Conclusions

Please consider better explaining the long-term risks over the years.

Author Response

COMMENT

Lines 27-28: Values expressed are mean ± standard deviation or uncertainty? Please explain.

RESPONSE

Since the standard deviation is the best physical estimate of the error on the mean value, only the mean values are expressed in the form: mean ± standard deviation. This is because it is the average value of the uncertainties of the individual measurements.

Materials and Methods

COMMENT

2.2.3. Please explain how or why Markus 10 doesn't wait until secular equilibrium.

RESPONSE

In the experimental protocol of the Markus 10 detector, no information was given about the secular equilibrium. In our opinion, the gas pumped into the soil does not escape during the measurement. Thus, we suppose and assume that all radioactive elements in the soil matrix involved are assumed to be in their equilibrium state.

Introduction

COMMENT

RESPONSE

Line 63: please replace thickness with depth

RÉPONSE

We agree with the reviewer.  The changes have been made in the revised manuscript. Line 63.

Materials and Methods

COMMENT

Line 92: Please check the size: 1 µm or 1 mm?

RESPONSE

The size is 1 mm. The correction was made in the revised manuscript.

COMMENT

Line 102: Check superscript (137C)

RESPONSE

The exponent has been readjusted in the revised manuscript.

COMMENT

Line 105: Check superscript (228Ac) and energy (usually 911 keV but 969 keV could be correct).

RESPONSE

The exponent has been readjusted in the revised manuscript.

COMMENT

Lines 108-110: Express units to counting time, declare type of counts.

RESPONSE

We agree with the reviewer. Corrections have been made in the revised manuscript. COMMENT

Eq. 2: Please check carefully: shouldn't the addends be squared?

RESPONSE

We agree with the reviewer, the corrections have been made in the revised manuscript. 

COMMENT

Lines 122-128: consider writing in a more technical, less “commercial” way.

RESPONSE

We agree with the reviewer, the corrections have been made in the revised manuscript.

Line 127-132

COMMENT

Check Eq.4 everything corrects in square brackets?

RESPONSE

Thanks for this remark. Indeed, Fb (0.98), the impact factor of the building material obtained experimentally on the site, is missing. The correction has been made in the revised manuscript. Lines 202-203

COMMENT

Eq. 9: Please check if AK/300 or AK/3000

RESPONSE

Indeed, it is AK/3000. Correction has been made in the revised manuscript.

COMMENT

Lines 227-240: summarize this information because Table 2 already shows all the data.

RESPONSE

This information has been summarized in the revised manuscript. Lines.299-306

COMMENT

Line 281: 68 or 53 kBqm-3

RESPONSE

This is the value of 53 kBq m-3. It represents the median value of radon concentrations in soil (Table 2). The correction was made in the revised manuscript.

COMMENT

Figure 3. Missing units for Rn222.

Figure 5. Units are missing.

RESPONSE

We agree with the reviewer. The units have been plotted on Fig.3 and Fig.5 in the revised manuscript.

COMMENT

Line 307: please, check if the sentence is finished.

RESPONSE

Indeed, the sentence is not finished. Corrections have been made, and the sentence has been completed in the revised manuscript. Lines...441-444

COMMENT

Lines 332-334: Please, review the text.

RESPONSE

The text has been revised and significant changes have been made to the revised manuscript. Lines 443-445.

COMMENT

Lines 343-387: please carefully check all numerical values and units both in the text and in table 4.

RESPONSE

Indeed, some of the data in the text were not consistent with those in Table 4. They were reviewed and significant changes were made to the revised manuscript. Lines 365-3744.

COMMENT

Figure 6. Consider if it makes sense to represent up to 1000 years.

RESPONSE

We agree with the reviewer. 1000 years is not a realistic time frame. Scientifically, presenting the risk out to 1000 years does not have a major impact. If one were to use the average life expectancy, one could limit oneself to representing the risk to 100 years. Nevertheless, the discussion of risk changes has focused on the zero-to-100-year range. Lines 535-572.

COMMENT

Table 6 and Figure 7 could be repetitive.

RESPONSE

We agree with the Reviewer. Table 6 has been improved to take into account of this comment.

Reviewer 3 Report

Report on “Correlation between ground 222Rn and 226Ra and long-term risk 2 assessment at the bauxite bearing area of Fongo-Tongo, west- 3 ern-Cameroon”

Written by Léonard Boris Djeufack et al.

I recommend this ms for publication after some minor corrections.

Delete word “Indeed” in all appearance.

Do not use word “daughter”. Replace with “progeny”.

Do not use Radium equivalent Raeq. It is already obsolete variable and not in use anymore.  Delete all about Raeq

Fig 3. Use units in ordinate axis.

Try to prepare text so that tables are on one page.

Author Response

COMMENT

Try to prepare text so that tables are on one page.

RESPONSE

Thanks for the comment. The long-term risk has been better explained in the conclusion of the revised manuscript. Line 705-718

COMMENT

Delete word “Indeed” in all appearance.

Do not use word “daughter”. Replace with “progeny”.

RESPONSE

The word "Indeed" has been deleted and the word "daughter" replaced by " progeny ". The corrections were made in the revised manuscript.

COMMENT

Do not use Radium equivalent Raeq. It is already obsolete variable and not in use anymore.  Delete all about Raeq

RESPONSE

We agree with the reviewer, and the recommendation has been addressed in the revised manuscript. The equivalent of Raeq has been removed from the text.

COMMENT

Fig 3. Use units in ordinate axis.

RESPONSE

Fig.3 has been redone with the units on the ordinate axis. The changes have been carried over into the revised manuscript.

COMMENT

Try to prepare text so that tables are on one page.

RESPONSE

The comment has been taken into account in the revised manuscript.